# Treatment of Diabetic Neuropathy with A Novel PAR1-Targeting Molecule

**DOI:** 10.3390/biom10111552

**Published:** 2020-11-13

**Authors:** Efrat Shavit-Stein, Shany Guly Gofrit, Alexandra Gayster, Yotam Teldan, Ariel Ron, Eiman Abu Bandora, Valery Golderman, Orna Gera, Sagi Harnof, Joab Chapman, Amir Dori

**Affiliations:** 1Department of Neurology, Sackler Faculty of Medicine, Tel Aviv University, Tel Aviv 6997801, Israel; valery.rodionov@gmail.com (V.G.); joabchapman@gmail.com (J.C.); amir.dori@gmail.com (A.D.); 2Department of Neurology, The Chaim Sheba Medical Center, Ramat Gan 5266202, Israel; shanygo@gmail.com (S.G.G.); alexisgenim@gmail.com (A.G.); teldanyotam@gmail.com (Y.T.); arielrona@gmail.com (A.R.); eiman.abu.bandora@gmail.com (E.A.B.); ornagera@gmail.com (O.G.); 3Department of Physiology and Pharmacology, Sackler Faculty of Medicine, Tel Aviv University, Tel Aviv 6997801, Israel; 4Department of Physical Therapy, Sackler Faculty of Medicine, Tel Aviv University, Tel Aviv 6997801, Israel; 5Department of Neurosurgery, Rabin Medical Center, Sackler Faculty of Medicine, Tel Aviv University, Tel Aviv 6997801, Israel; sharnof@gmail.com; 6Robert and Martha Harden Chair in Mental and Neurological Diseases, Sackler Faculty of Medicine, Tel Aviv University, Tel Aviv 6997801, Israel; 7Talpiot Medical Leadership Program, The Chaim Sheba Medical Center, Ramat Gan 5266202, Israel

**Keywords:** thrombin, PAR1, diabetes mellitus, streptozotocin, peripheral neuropathy

## Abstract

Diabetic peripheral neuropathy (DPN) is a disabling common complication of diabetes mellitus (DM). Thrombin, a coagulation factor, is increased in DM and affects nerve function via its G-protein coupled protease activated receptor 1 (PAR1). Methods: A novel PAR1 modulator (PARIN5) was designed based on the thrombin PAR1 recognition site. Coagulation, motor and sensory function and small fiber loss were evaluated by employing the murine streptozotocin diabetes model. Results: PARIN5 showed a safe coagulation profile and showed no significant effect on weight or glucose levels. Diabetic mice spent shorter time on the rotarod (*p* < 0.001), and had hypoalgesia (*p* < 0.05), slow conduction velocity (*p* < 0.0001) and reduced skin innervation (*p* < 0.0001). Treatment with PARIN5 significantly improved rotarod performance (*p* < 0.05), normalized hypoalgesia (*p* < 0.05), attenuated slowing of nerve conduction velocity (*p* < 0.05) and improved skin innervation (*p* <0.0001). Conclusion: PARIN5 is a novel pharmacological approach for prevention of DPN development, via PAR1 pathway modulation.

## 1. Introduction

Diabetes mellitus (DM) is a chronic disease affecting more than 285 million individuals worldwide [1]. Among diabetic patients, 60% develop diabetic peripheral neuropathy (DPN) [2]. DPN includes sensory neuropathy, with both hyposensitivity and hyperalgesia, and early involvement of small fibers [3]. Less commonly, or later in the course of the disease, motor fibers are affected [4]. The best current treatment for DPN is prevention by tight glycemic control [4] and symptomatic relief. The high prevalence of this complication and the absence of effective therapy emphasize the need for novel and targeted drug development.

Thrombin is a well-known key coagulation factor but additionally participates in cellular processes including systemic inflammation [5], neural inflammation [6] and neuronal degeneration [7]. Thrombin and its G-protein coupled protease activated receptor 1 (PAR1) were previously identified at the node of Ranvier of peripheral nerves. Accordingly, activation of the thrombin pathway caused a reversible conduction block [8]. Thrombin levels are increased in peripheral nerves of diabetic mice [9] and in the blood of diabetic patients, where high levels correlate with poor glycemic control [10]. Taken together, thrombin’s influence on peripheral nerve function and its increase in diabetes suggest it as an important player in the pathogenies of DPN. Inhibition of the thrombin-activated PAR1 pathway and downstream negative cellular effects may thus be a target for intervention.

The use of thrombin modulators aiming at improving nerve conduction may hold some risks due to their anti-coagulation effects [9]. We presently developed a specific thrombin/PAR1-activating modulator using a novel approach based on the PAR1 activation site, which spares anti-coagulation side-effects. This molecule is composed of five amino acids derived from the PAR1 sequence, and thus was named PARIN5.

In the present study, we tested the neuroprotective effect of PARIN5 in the in vivo streptozotocin (STZ) mice model for diabetic neuropathy. We found that treatment with PARIN5 prevented development of thermal sensory dysfunction in diabetic mice, alongside preservation of intradermal nerve fibers density. Similarly, motor function and nerve conduction velocity in diabetic animals were protected by PARIN5 treatment.

## 2. Materials and Methods

### 2.1. Molecule Design

The PARIN5 molecule was designed according to the thrombin-specific recognition site sequence of the PAR1 N-terminal (^35^NATLDPR^41^) and includes a 5-amino acid backbone. The molecule is protected by a tosyl group at the N-terminal and conjugated to a serine active site blocker, chloro-methyl-ketone (CMK), at the C-terminus. The length of the backbone molecule was determined in preliminary studies according to highest efficacy together with safety profile. The molecule was synthesized by the American Peptide Company (Sunnyvale, CA, USA), with over 96% purity grade verified by HPLC.

### 2.2. Ethical Considerations

Experiments were approved by the Sheba Medical Center Animal Welfare Committee (920/14/ANIM) and appropriate measures to avert pain and suffering to the animals were taken. Male C57BL/6, 8-week-old mice (Envigo, Israel) were maintained in a controlled facility at 18–22 ° C and 40–60% humidity, with 12 h dark/12 h light cycles and allowed free access to water and food.

### 2.3. Induction of Diabetes

Diabetes was induced by a single (150 mg/kg) intraperitoneal (IP) injection of streptozotocin (STZ, S0130, Sigma-Aldrich, St. Louis, MO, USA, 0.1M sodium citrate buffer, pH 4.5). Glucose levels were measured by a commercial glucometer device on three repeated occasions within a week from STZ injection and mice were considered hyperglycemic when glucose levels were above 250 mg/dl. Body weight was measured daily (Figure 1).

### 2.4. Treatment Protocols

Animals were allocated into 4 groups: (1) control healthy mice, (2) diabetic mice and (3,4) diabetic mice treated with PARIN5 dissolved in saline used as a vehicle, at low (37.5 ng/kg, 10nM) or high (375 ng/kg, 100 nM) doses, respectively. Sensory and motor evaluation was conducted at days 30 and 42 following STZ injection, respectively. Nerve conduction studies and small fiber density counting were conducted following 35 days of treatment. PARIN5 treatments and control saline were administered by IP injection 5 times a week, starting 2 weeks following STZ injection.

### 2.5. Coagulation in Plasma

HemosIL normal control (Instrumentation Laboratory) human pooled citrated plasma from healthy donors was used as control for coagulation studies. Lyophilized human plasma containing buffer, stabilizers and preservatives was reconstituted using standard laboratory methods. Plasma was exposed in vitro to the indicated concentrations of PARIN5 for 10 min. Prothrombin time (PT), activated partial thromboplastin time (aPTT) and thrombin time (TT) were then measured. All tests employed an ACLTOP^®^ 500 autoanalyzer at the Sheba Medical Center (Ramat Gan, Israel) MegaLab according to standard operating procedures (SOP). Results represent an average of 5 measures. The lowest PARIN5 dose used was 10^−10^ M, which was equivalent to the saline negative control.

### 2.6. Motor Function Evaluation

Motor performance was assessed by means of a rotarod test. Mice were pre-trained to run on the rod which rotated at a fixed speed of 19 rounds per minute. After induction, mice were assessed at day 42. Mice were allowed to run for up to 60 s on each trial, or until they fell off the rod. The mean of three consecutive trials was recorded for each animal.

### 2.7. Sensory Function Evaluation

Sensory function was evaluated by thermal response on a hot plate at day 30 following STZ injection. Animals were placed in a perspex cylinder on a heated stage maintained at 50 ± 1° C. Time to heat response indicated by hind paw licking, shaking or jumping was measured. A maximum on-plate time was set to 30 s to prevent skin injury.

### 2.8. Nerve Conduction Measurements

Nerve conduction studies were performed at day 43, with a standard electrodiagnostic medicine (Viking, Natus) device. Animals were anesthetized with ketamine (100 mg/kg) and xylazine (10 mg/kg) mixed solution before recordings. Sciatic nerve electrophysiological studies were conducted in the prone position. Compound muscle action potential (CMAP) responses were recorded from the gastrocnemius muscle with an active needle electrode and a reference electrode placed at the center of the foot. Stimulation of the sciatic nerve was performed at the ischial notch with a blunt cathode needle electrode. The anode was placed 5 mm proximal to the cathode and the ground electrode was placed between the stimulating and recording electrodes. Supra-maximal stimulation, at a range of 3–5 mA, was employed, with low- and high-frequency filters set to 10 Hz and 10 kHz, respectively. To calculate the motor nerve conduction velocity (MNCV), the distance between the stimulation and recording sites was divided by the latency. The latency of neuromuscular junction activation was not subtracted. CMAP amplitudes were measured from the baseline to the negative peak.

### 2.9. Intraepidermal Nerve Fiber Density (IENFD) Measurement in Skin

Hind paw plantar skin biopsies harvested on day 43 following STZ injection were collected, fixed and frozen as detailed above. Cryosections (14 µm thick) were adhered to charged slides, air-dried, rinsed with 0.1M TBS solution and incubated in blocking solution (0.1M TBS, 1% skim milk powder, 10% NHS and 10% Triton X-100) for 4 h at RT and incubated with the primary antibody (anti-PGP9.5, C-Terminal cat# SAB4503057, Sigma (1:400) in 1:1 diluted blocking solution/0.1M TBS) at 4 °C overnight. Slides were washed with TBS and incubated with secondary antibody (488 conjugated Alexa goat anti-rabbit 1:100 in 1:1 diluted blocking solution/0.1M TBS) for 4 h in a dark room at RT and covered with FluromountTM (Diadnostic Biosystem cat#K024–25mL). Sections from each animal were randomly analyzed using an Olympus microscope and the IENFD, expressed as the average number of nerve fibers per millimeter epidermis, quantified according to guidelines published by the European Federation of Neurological Societies [11].

### 2.10. Statistics

Data analyses and assessments were performed using GraphPad software (GraphPad Prism 6 for Windows, GraphPad Software, La Jolla, CA, USA).

Weights and glucose levels were assessed using mixed-effect analyses with repeated measures prospectively comparing each treatment group with untreated diabetic animals and normal controls. Comparisons between continuous variables (sensory and motor functions, nerve conduction) were calculated using one-way ANOVA with corrections for multiple comparisons. IENFD was evaluated by two independent observers with the kappa coefficient calculated for inter-rater agreement.

All illustrations were created using Biorender.com.

## 3. Results

### 3.1. Coagulation Studies

The effect of PARIN5 on coagulation was evaluated by PT, PTT and TT studies. Various concentrations of PARIN5 were tested to identify a favorable safe dosage for systemic administration. PARIN5 had no effect on PT, PTT or TT up to a concentration of 10^−6^M (Figure 2). Accordingly, we determined a safe low dose (37.5 ng/kg, 10 nM) and high dose (375 ng/kg, 100 nM) for further in vivo studies.

### 3.2. Weight and Blood Glucose Levels

Diabetes developed in over 70% of STZ-injected mice (n = 26/36), showing blood glucose levels above 250 mg/dL. Animals that did not show diabetes were excluded from the study. PARIN5 treatments did not influence blood glucose levels (F(2,33) = 0.87, *p* = 0.42 for treatments, Appendix A). Diabetic mice showed a significant decrease in body weight compared to controls, starting on day 10 (18.8 ± 0.97, 23.24 ± 0.44 gr, *p* = 0.01) and throughout the remaining study period (*p* < 0.0001, F(3,25) = 11.08). Weight was not significantly affected by PARIN5 treatments compared to non-treated diabetic mice (F(2,15) = 0.06, *p* = 0.93 for treatments, Appendix A).

### 3.3. Motor Function

Motor function evaluated by the rotarod test was measured on day 42. Diabetic animals showed a significant decrease in performance compared to healthy controls (15.8 ± 5.8, 50.3 ± 4.1 sec, respectively, *p* < 0.001, Figure 3A). PARIN5 treatments attenuated motor function impairment of diabetic mice with a dose-dependent response, which was significant with the higher dose (375 ng/kg) used (Figure 3A). Animals treated with high-dose PARIN5 (375 ng/kg) showed no significant decline in motor impairment in comparison to controls (*p* = 0.19) and performed significantly better than untreated diabetic mice (*p* < 0.05).

### 3.4. Sensory Function

Sensitivity to heat was determined by measuring the duration (latency) of response to high temperature on a hot plate at day 30. High values represent hyposensitivity and low values indicate hypersensitivity. Diabetic mice showed hyposensitivity to heat compared to controls (latencies of 24.6 ± 2.2, 18.1 ± 1.2 sec, *p* < 0.05, Figure 3B). PARIN5 treatments prevented development of hyposensitivity in a dose-dependent manner. High-dose PARIN5 (375 ng/kg) treatment prevented development of hyposensitivity to heat and showed a response similar to controls (latency of 16.7 ± 2.4 sec; *p* > 0.86), which was significantly different in comparison to untreated diabetic mice (*p* < 0.05, Figure 3B).

### 3.5. Nerve Conduction Studies

Nerve conduction velocity was significantly slower in diabetic mice compared to controls (26.2 ± 4.3, 51.6 ± 2.2 m/sec, *p* < 0.0001, Figure 4A), while CMAPs were not significantly affected (3.01±0.6, 3.8 ± 0.6 mV, *p* = 0.76, Figure 4B). This suggests that diabetes disrupted the large fiber myelin structure but did not cause axonal damage at this relatively short interval. PARIN5 high-dose (375 ng/kg) treatment significantly attenuated the decrease in conduction velocity compare to untreated diabetic mice (39.9 ± 4.4 m/sec, *p* < 0.05, Figure 4A). Treatments did not show a significant effect on CMAP amplitude (3.0 ± 0.4, 2.4 ± 0.5 mV for PARIN5 low (37.5 ng/kg) and high (375 ng/kg) doses, respectively, *p* > 0.05, Figure 4B).

### 3.6. Intradermal Nerve Fiber Density

Diabetic mice showed a significant reduction in skin innervation as determined by IENFD compared to controls (4.1 ± 0.4, 10.3 ± 0.5, respectively, *p* < 0.0001, Figure 5). This indicates that the small (unmyelinated sensory) fiber loss as detected in skin precedes the loss of large (motor) axons determined by CMAP amplitude, consistent with small fiber neuropathy which occurs early in the course of diabetic polyneuropathy [3]. Treatment with high-dose PARIN5 (375 ng/kg) significantly attenuated the loss of these small axons from skin compared to untreated diabetic mice, to almost complete normalization (9.4 ± 0.7, *p* < 0.0001, Figure 4).

## 4. Discussion

In the present study, we employed PARIN5, a novel selective PAR1 inhibitor, to modulate the effect of increased thrombin on peripheral nerves in the mouse STZ model for diabetic neuropathy. PARIN5 treatment preserved motor and sensory function, sciatic nerve conduction velocity and epidermal small fiber density in diabetic mice. This indicates a protective effect against development of DPN and supports the involvement of the thrombin/PAR1 pathway in the pathogenesis of diabetic neuropathy. Coagulation studies were only minimally affected by the dose range of PARIN5 used in the present study, suggesting that the beneficial effect of PARIN5 is not due to anti-coagulation properties, but rather due to the cellular neuromodulation effects.

The multifactorial G-protein coupled receptor PAR1 is a known crossroad molecule in nerve conduction, structure and function in both the peripheral and central nervous systems (PNS and CNS, respectively) [12]. Its modulation is highly complex and has the potential to either damage or protect neurons and glia cells. Accumulated recent data point to the crucial role the thrombin/PAR1 system plays in the PNS function and the fine neuron–glial interactions.

PAR1 is located at the node of Ranvier [8], a key structure for nerve conduction. Activation of PAR1 by high thrombin levels inhibits nerve conduction [8] and leads to the destruction of nodal morphology [13]. These effects are prevented by using thrombin and PAR1 inhibitors. High thrombin levels and PAR1 activation with an agonist peptide cause Schwann cell changes, including demyelination of the paranode region [14], and induce motor neuron cell death in ALS models [15]. However, low thrombin levels induce PAR1 Schwann cell modification to support neuronal survival [14]. Interestingly, PAR1 activation by a different coagulation factor, activated protein C (aPC), induces neuronal differentiation [16], enhances LTP in the CNS [17] and promotes a neuroprotective phenotype in Schwann cells in the PNS [18]. Together, these points strongly support the idea that the PAR1 pathway is an important target for fine-tuning pharmacological manipulation in neurological diseases. Diabetes causes disruption of the peripheral nerve’s node of Ranvier [19], which is consistent with the slowing of nerve conduction velocity [20]. We previously reported the inhibition of PAR1 stimulation by thrombin-preserved node of Ranvier morphology in diabetic rats [9], indicating that selective PAR1 inhibition may be an important target for pharmacological intervention. We chose to modulate this pathway using the specific N-terminal PAR1 sequence recognized by thrombin as a template for the unique PARIN5 molecule, aiming to specifically inhibit PAR1 activation by thrombin. The PARIN5 molecule was previously designed as part of a set of molecules which differ in their backbone length, based on the thrombin recognition site sequence of PAR1, and found to be potent PAR1 inhibitors [21]. Based on the specificity of the protease-binding site on PAR1 in comparison to PAR3 and PAR4, it is reasonable to attribute the specificity of PARIN5 to PAR1. Though our unpublished antibody staining and RT-PCR studies indicate a very low presence of PAR4 in the sciatic nerve, a beneficial effect of PARIN5 due to PAR4 presence on platelets and the vascular endothelium cannot be excluded.

Neuropathy is common in many systemic diseases and can be found in autoimmune disorders [22,23], chronic infections [24], metabolic disorders [25] and as medication side-effects [26]. Diabetic neuropathy was chosen in the current study as a representative neuropathy for the evaluation of PARIN5 due to multiple reasons. DM is extremely common, and DPN affects the majority of DM patients [2]. DPN has a large impact on quality of life. Especially disturbing is the presence of painful diabetic neuropathy, characterized by hyperalgesia [4,27] and formication. The STZ injection rodent DM model, which has the benefits of being both easily established and reliably monitored by blood glucose measurements, presents with hyperalgesia reminiscent of that of patients [27]. Furthermore, this model previously showed elevated levels of thrombin in the sciatic nerve, disruption of the node of Ranvier and decreased conduction velocity, which were prevented by thrombin inhibition [9,14]. Indeed, our results in the untreated DM mice indicate slower nerve conduction velocities, decreased motor function and the expected hyperalgesia which is a common characteristic for diabetic patients. The latter is accompanied by a reduction in small fiber density in skin biopsies, which was previously described in patients with painful sensory neuropathies [28].

This study has some limitations. The focus of the present study is diabetic neuropathy, which is merely one of many. The influence of PAR1-modulating treatments in other models for neuropathy remains to be investigated. We previously showed that thrombin activity is significantly elevated in the sciatic nerve of STZ-injected animals [9], therefore this measure was not repeated in this study. The effect of PARIN5 treatment on thrombin activity levels in the sciatic nerve at several time points may be addressed in a future study, shedding light on the temporal resolution of the thrombin/PAR1 destructive processes and PARIN5 protection.

The source of the elevated thrombin activity in the sciatic nerve associated with deleterious PAR1 activation may be due to local overproduction or systemically elevated thrombin levels together with blood–nerve barrier leakage due to diabetic vasculopathy, or both. Further study is needed for the evaluation of PARIN5 effects on the vascular component of diabetic neuropathy. This may have consequences on future indications and treatment modes of administration.

The STZ mouse has an elevated tendency for infections, resembling DM patients [29]. Therefore, the IP injection mode of administration may lead to further complications. Since PARIN5 treatment selectively improved neural function, with no effect on general diabetes measurements such as blood glucose levels or body weight, there is a possible advantage for the local mode of administration.

PAR1 is an important receptor in the coagulation system. Its inhibition has the potential to interfere with coagulation functions. In this study, we treated the DM mice by repeated PARIN5 administrations. The CMK modification in the PARIN5 molecule irreversibly inhibits the protease active site. Although the results of the coagulation studies suggest a relatively large therapeutic window, the cumulative effect of PARIN5 is not clear. PARIN5 is composed of a modified peptide of a five-amino acids chain. Peptide-based drugs are prone to enzymatic degradation and decreased activity and altered versions of PARIN5 may be more efficient. Furthermore, the effect of different metabolisms and other pharmacokinetics parameters in different species require further evaluation, as well as different timings of treatment, and perhaps consideration of a preventive treatment protocol. Diabetic patients have altered coagulation, manifested as hypercoagulability, due to the dysfunction of the endothelium, platelets and coagulation factors [30]. Although PARIN5 at doses used in the present study had no effect on the coagulation studies, its possible anti-coagulation effect may show additional benefit in reducing overactivation of coagulation in diabetic patients.

## 5. Conclusions

The present study describes a novel pharmacological approach for prevention of DPN by modulation of the PAR1 pathway. This treatment prevented development of sensory and motor dysfunction and preserved sciatic large fiber nerve conduction and small fiber density in skin. This pathway opens a new avenue for the treatment of a debilitating common peripheral nerve disease.

## 6. Patents

A patent application based on these findings has been granted (# 10028999).

## Figures and Tables

**Figure 1 biomolecules-10-01552-f001:**
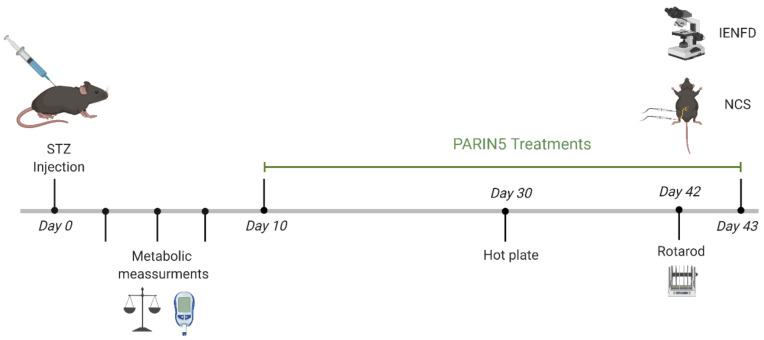
Timeline.

**Figure 2 biomolecules-10-01552-f002:**
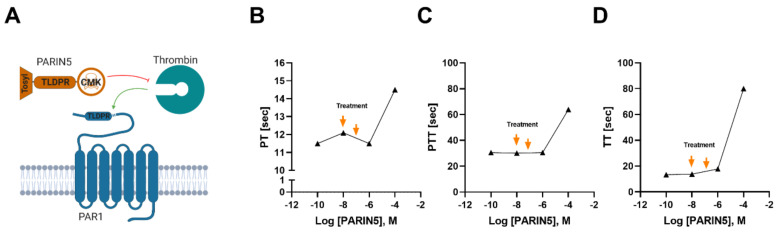
Molecular design and coagulation studies: (**A**) PARIN5 is based on sequence homology with PAR1, which makes it recognized by thrombin. The molecule was modified with a tosyl group as a protective group at the N-terminal, and with chloro-methyl-ketone (CMK) as a serine active site blocker at the C-terminal. Coagulation studies (**B**–**D**) were conducted to identify the optimal therapeutic window for PARIN5 treatment. Orange arrows correspond to treatment dosages.

**Figure 3 biomolecules-10-01552-f003:**
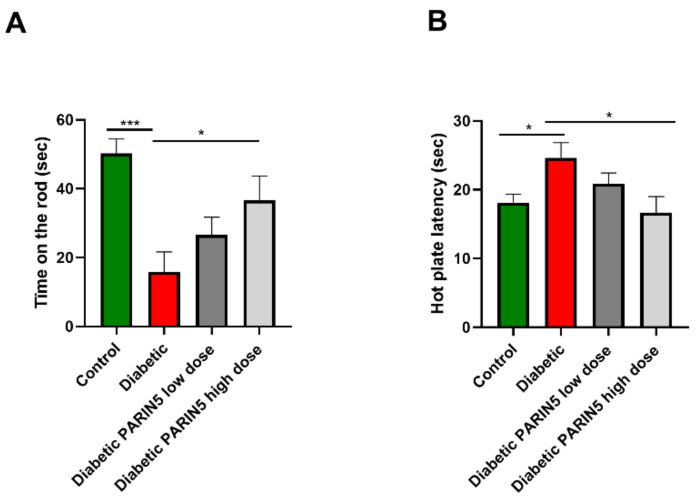
Motor and sensory evaluation: (**A**) Motor evaluation by the rotarod test. Diabetic mice showed shorter duration time on the rod, suggesting motor dysfunction. Motor performance was significantly improved by PARIN5 high-dose (375 ng/kg) treatment. (**B**) Diabetic mice were hyposensitive to temperature as evaluated by the hot plate test. Hyposensitivity was significantly improved following PARIN5 high-dose (375 ng/kg) treatment. Results are presented as mean ± SEM, * *p* < 0.05, *** *p* < 0.001.

**Figure 4 biomolecules-10-01552-f004:**
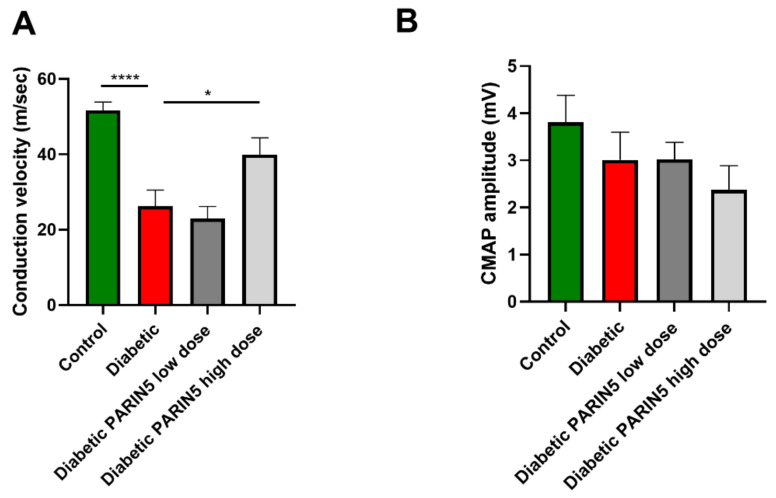
Nerve conduction deficits prevented by PARIN5 treatment. (**A**) Nerve conduction velocity was significantly slower in diabetic mice and was significantly higher following PARIN5 treatment. (**B**) Amplitude was not changed in diabetic mice compared to control and was not affected by treatments. Results are presented as mean ± SEM, * *p* < 0.05, **** *p* < 0.0001.

**Figure 5 biomolecules-10-01552-f005:**
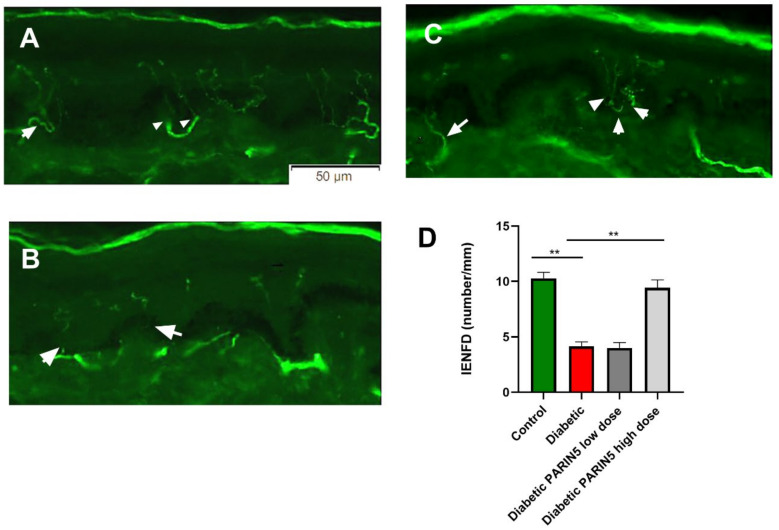
Intraepidermal nerve fiber density (IENFD), skin biopsy images and quantification: representative figures of skin biopsies taken from (**A**) a healthy control mouse, (**B**) a diabetic mouse and (**C**) a diabetic mouse treated with high-dose (375 ng/kg) PARIN5. White arrowheads mark small fibers. Small fibers are barely seen crossing the dermal epidermal junction in the diabetic mice. (**D**) Quantification of small fibers density in mice from all groups showed significantly decreased IENFD in diabetic mice compared to control. PARIN5 high-dose (375 ng/kg) treatment significantly improved IENFD. Results are presented as mean ± SEM, ** *p* < 0.01.

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
