# Peer review of "Treatment of Diabetic Neuropathy with A Novel PAR1-Targeting Molecule"

_biomolecules, 2020, doi:10.3390/biom10111552_

Round 1

Reviewer 1 Report

This study investigates the therapeutic potential of a new PAR1 inhibitor (PARIN5) in the setting of diabetic neuropathy. The authors have selected the thrombin specific recognition site sequence of the PAR1 N-terminal (35NATLDPR41) to design their 5-amino acid PAR1 inhibitor. The administration of PARIN5 to streptozotocin (STZ)-induced diabetic mice ameliorates several features of diabetic neuropathy, without changes in metabolic parameters (blood glucose levels and body weight). The hypothesis is clearly presented and the manuscript is well written. Overall, this is an interesting study that may provide additional insights into the role of thrombin-PAR1 in diabetic complications.

Below are some concerns that should be addressed to improve the message.

Major

- This study is based on the assumption that thrombin levels are increased in the diabetic mouse model used. If the authors are unable to obtain these data, perhaps this can be mentioned as a limitation of the study.

- What is the evidence that PARIN5 inhibits thrombin binding to PAR1?

- What is the specificity of PARIN5 to PAR1 in comparison to other PAR isoforms (e.g., PAR3 and PAR4)? Could this small peptide also bind to other targets, besides PAR isoforms?

- Coagulation studies conducted in plasma from healthy controls showed that PARIN5 (10 nM and 100 nM) did not affect any of the parameters tested. The authors should discuss the possibility that the results could have been different if the plasma was obtained from diabetic patients, who have an altered coagulation profile.

- The beneficial effects of PARIN5 on several features of diabetic peripheral neuropathy seem to be independent of its effects on coagulation and metabolism. Can the authors provide some mechanistic explanation for their findings in the Discussion section?

Minor

- Glucose levels are shown until day 7, whereas body weight measurements are reported until day 43. Could the authors provide additional time points for glucose levels?

- In order to make the legend for Figure 2 clearer, there should be an indication in the legend that the two yellow arrows correspond to PARIN5 treatment (10 nM and 100 nM).

Author Response

Below are some concerns that should be addressed to improve the message.

Major

- This study is based on the assumption that thrombin levels are increased in the diabetic mouse model used. If the authors are unable to obtain these data, perhaps this can be mentioned as a limitation of the study.

We thank the reviewer for this comment. Thrombin activity levels in the diabetic sciatic nerve were recently evaluated by our group and published (Shavit-Stein et al. 2019, the role of thrombin in the pathogenesis of diabetic neuropathy, Plos one). Therefore, in this current manuscript the measurements were not repeated. Indeed, measuring thrombin levels in the sciatic nerve in response to the treatment was not performed here, is complex (temporal effect etc.), and should be addressed in a dedicated future study. This was added as a limitation to the discussion section (pages 9).

- What is the evidence that PARIN5 inhibits thrombin binding to PAR1?

This is a highly important comment. In a previous work by our group (Shavit-Stein et al. 2018, A Novel Compound Targeting Protease Receptor 1 Activators for the Treatment of Glioblastoma, Frontiers in Neurology, Figure 1C), we designed a set of molecules that bind PAR1 based on the recognition site of thrombin on PAR1. The molecules differ by the length of their backbone, and all of them showed inhibition of thrombin activity. Their inhibition of PAR1 activation was further supported in vitro by the reduction of pERK levels in response to thrombin activation in comparison to control. pERK is a known downstream signaling molecule indicating PAR1 activation. A comment regarding the design of PARIN5 and its ability to inhibit PAR1 activation by thrombin was added to the discussion section (page 9).

- What is the specificity of PARIN5 to PAR1 in comparison to other PAR isoforms (e.g., PAR3 and PAR4)? Could this small peptide also bind to other targets, besides PAR isoforms?

This is indeed an important but complex question. The specificity of the effect may be dependent on both the receptors and the relatively large number of proteases that activate them. Thrombin binds PAR upstream to the cleavage site in a recognition sequence composed of AA with the highest affinity to the first AA upstream to the cleavage site and decreased affinity at more distant from the cleavage site. The PAR1 cleavage site is between R41 and S42 and the recognition sequence is   26ARRPESKATNATLDPR41where 37TLDPR41 is the 5AA with the highest affinity chosen as the basic backbone sequence for PARIN5. The corresponding PAR3 sequence is 34TLPIK38, and PAR4 corresponding sequence is 43LPAPR47. A general comparison of these sequences indicates 40% similarity between the recognition sequence of PAR1, PAR3 and PAR4. Since the highest affinity is attributed to the closest AA to the cleavage site, we believe that there is a higher possibility of PAR4 inhibition by PARIN5, rather than PAR3. We have in the past stained sciatic nerves with antibodies for both PAR3 and PAR4 and have not found any neuronal binding while RT-PCR results indicated some PAR3 but very little PAR4 expression in the nerves (unpublished). However, since PAR4 receptor is present mainly on platelets and vasculature cells (leukocytes, endothelial, smooth muscle) rather than neural cells we believe that if indeed PARIN5 inhibits PAR4 this may contribute to the protective effect in this diabetic neuropathy model. Another issue is that there are several different proteases known to activate PAR1 and not PAR3 and PAR4 (in addition to thrombin) so the downstream effects associated with PARIN5 treatment are most likely due to specific PAR1 inhibition. A brief discussion of these issues has been added to the Discussion (page 9).

- Coagulation studies conducted in plasma from healthy controls showed that PARIN5 (10 nM and 100 nM) did not affect any of the parameters tested. The authors should discuss the possibility that the results could have been different if the plasma was obtained from diabetic patients, who have an altered coagulation profile.

We thank the reviewer for this important comment. Indeed, diabetic patients present altered coagulation functions. A comment regarding the potential inhibitory effect of PARIN5 in DM patients was added to the discussion section (page 10).

- The beneficial effects of PARIN5 on several features of diabetic peripheral neuropathy seem to be independent of its effects on coagulation and metabolism. Can the authors provide some mechanistic explanation for their findings in the Discussion section?

An explanatory comment regarding the cellular, rather than anti-coagulation, beneficial effects of PARIN5 was added to the beginning of the discussion section (page 8), followed by an explanation regarding the cellular role of PAR1.

Minor

- Glucose levels are shown until day 7, whereas body weight measurements are reported until day 43. Could the authors provide additional time points for glucose levels?

Additional information regarding glucose levels later in the course of the disease was added to the supplementary figure.

- In order to make the legend for Figure 2 clearer, there should be an indication in the legend that the two yellow arrows correspond to PARIN5 treatment (10 nM and 100 nM).

This was added to the legend.

Reviewer 2 Report

The paper was executed to study neuroprotective effect of PARIN5, PAR1 targeting molecule, in murine streptozotocin diabetic model. The authors concluded that PARIN5 is involved in preventing DPN development through modulation of PAR1 pathway. The study is interesting but the following issues need attention.

1) Abstract

Introduction and method are too long. The results are truncated.

2) The data supporting the conclusion are not complete. Data are rough.

3) Coagulation studies

Any data with PARIN5 vehicle, or negative(positive) control?

4) The doses of PARIN5 are needed to include in the text.

5) Skin biopsy images are not clear.

Author Response

The paper was executed to study neuroprotective effect of PARIN5, PAR1 targeting molecule, in murine streptozotocin diabetic model. The authors concluded that PARIN5 is involved in preventing DPN development through modulation of PAR1 pathway. The study is interesting but the following issues need attention.

1) Abstract

Introduction and method are too long. The results are truncated.

Abstract was rephrased, introduction and methods were shortened and the lack of effect on glucose levels and body weight was added.

2) The data supporting the conclusion are not complete. Data are rough.

Additional data regarding glucose levels was added.

3) Coagulation studies

Any data with PARIN5 vehicle, or negative(positive) control?

Saline was used as a vehicle for PARIN5 in all experiments and therefore this was the control sham treatment. Coagulation studies were conducted in the Sheba Medical Center (Israel) MegaLab according to standard operating procedures (SOP). The ACLTOP® 500 autoanalyzer is calibrated each day with positive and negative controls. The last evaluated concentration of PARIN5 was 10-10M, which was found equivalent to saline alone. We added details of these issues to the results section (page 3).  

4) The doses of PARIN5 are needed to include in the text.

PARIN5 dose was added to the text.

5) Skin biopsy images are not clear.

We have attempted to improve the figure. The skin pictures obtained by fluorescence microscopy demonstrate only a portion of the small fiber in the epidermis, which are in the focus plain. To visualize and count the fibers we vary the focus of the objective throughout the superficial to deep layers of the section. This is in contrast to confocal microscopy, which stacks all the small fibers in the section to a flat picture.

Round 2

Reviewer 1 Report

The authors have answered satisfactorily to all my requests and improved the presentation of their results. 

Reviewer 2 Report

The authors attempted to revise their manuscipt according to the comments.